# Why Equity Follows the Law

**Adam J. MacLeod**

School of Law, St. Mary's University, San Antonio, TX 78228, USA; amacleod@stmarytx.edu

**Abstract:** Renewed attention to equity in higher education is welcome because true equity helps us to reason together well. When administered correctly, the jurisprudence of equity models civil discourse and, therefore, can teach us how to carry out civic engagement reasonably. Equitable interpretation of the law teaches us how to understand each other charitably. And equity's deference to law teaches us how to reason well together about our practical problems. Law is the practical reasoning that we do together. Equity serves the ends of justice by serving law, rather than undermining it. These functions of equity in adjudication point toward a model of equity in practical reasoning and civic discourse more broadly. Research method: jurisprudence.

**Keywords:** equity; law; justice; natural justice; legal justice; civic discourse; legal education

## 1. Introduction

Worthwhile civic engagement begins with sound practical reasoning. If civic engagement is to be modeled and carried out well, it must begin with reasoned discourse about the practical questions that our political communities confront each day. Unless we reason together, civic engagement threatens to devolve into interminable collisions between well-intended but mutually incompatible plans of action, or worse, raw assertions of power.

For this reason, renewed attention to the role of equity in civic life (as in initiatives focused on diversity, equity and inclusion) is welcome. Equity helps us to reason together well about fundamental questions of justice. But equity plays a positive role in civic affairs only if it is administered well. Equity works well when it facilitates productive practical reasoning but can be destructive if it attempts to displace practical reasoning with bare diktat. In particular, to be effective, equity must take a back seat to law.

Equity's charity toward and deference to law can teach us how to reason well together because and insofar as it follows law. Law is arguably the most important locus of civic discourse, and it is the primary institution by which we achieve some measure of justice in our civic affairs. Equity serves well by serving law, rather than undermining it. Correctly understood, equity takes legal rights as its starting point and operates to complete the law.

Law schools have a particularly important role to play in teaching equity and law together. Lawyers are fitted for their roles as civic leaders to the extent that they are educated in the norms and institutions that support the rule of law *and* in the principles and modes of reasoning that characterize an equitable interpretation of the law. This dual model of legal education—teaching legal justice and natural justice, law and equity—characterized legal education in Europe from the founding of the University of Bologna in the eleventh century and later in England and her North American colonies (Helmholz 2015). But lawyers and law faculties have largely neglected this tradition over the last century or so (George 1999; Banner 2021, pp. 167–250). Educators in law, jurisprudence and philosophy, politics, government and related disciplines would do well to recover it.

## 2. Essential Characteristics of Equity

When administered correctly, the jurisprudence of equity perfects legal justice. And the way in which equity serves the goal of justice supplies a model for productive civil

discourse. Equity jurisprudence can teach us how to deliberate well about the civic activities we carry out together.

Equity is not only a mode of reasoning; it is also a set of principles, a source of jurisdiction, a body of authoritative doctrines and, in some places, an institution of adjudication distinct from legal institutions. A concept of equity that encompasses equity's many aspects must, therefore, not attempt to be too precise. But for the purposes of this paper, we can stipulate a rough, working definition with attention to equity's function as a mode of practical reasoning. Here, equity is regarded as a way of reasoning about practical problems whose goal is to perfect law toward the end of achieving justice. As a way of reasoning about justice, equity displays certain essential characteristics. Equity's essential characteristics fit well with the requirements of practical reasoning in a community.

First of all, equity is charitable. In their equitable mode of interpretation, judges discern the "true sense of the law" (Blackstone 2016, p. 283), beginning with a presumption that legislatures intended what is reasonable when their posited rules are applied in particular cases (Aquinas 1964, pp. 342–46; MacLeod 2018b, pp. 124–26). The role of equity in adjudication is not merely to fill gaps in the law but rather to enable judges to presuppose, where the text allows, that the legislature intended the just result and to rule accordingly (Shiner 1994, pp. 1254–60). "Plainly stated, equity employs the hypothetical judgment of the legislator to correct the deficiency resulting from the law's universality" (Zahnd 1996, p. 267). Equity does what a just and reasonable legislator would have wanted done but could not have accomplished using the tools of positive law (Smith 2021, pp. 1068–69).

Second, equity is just. Within its limited jurisdiction, equity acts for reasons of natural right, as defined by the ancient principles and procedures laid down in Roman, canon and civil law, and in later common law (Story 1836, pp. 18–19; Haskett 1996, pp. 256–68). The classical concept of equity as the jurisdiction "founded in natural justice, in honest and right" was received early into Roman law and common law and plays a significant role in both the civil law and common law traditions (Home 2014, pp. 19–27; Story 1836, pp. 1, 4–6, 10–12; Zahnd 1996, pp. 270–75; Shanske 2005, pp. 2059–66). In both traditions, as in Aristotle's original account, equity and law complement each other insofar as each is an indispensable part of justice—legal justice and natural justice (Haines 1916, pp. 621–23; Aquinas 1964, pp. 344–45; Haskett 1996, p. 267).

Finally, equity presupposes what Aquinas called the "rectitude of law [and] of legal justice" and therefore defers to law (Aquinas 1964, p. 344). As jurists have long taken pains to insist, where the law is clear, courts of equity must follow the law in their judgments and decrees no less rigorously than courts of law do (Blackstone 2016, p. 283; Story 1836, pp. 12–13, 15). Though equity has the potential to undo the law in its application within particular cases, equity takes the law as given (Rundell 1958, p. 82). As Frederic Maitland famously expressed it, "Equity without common law would have been a castle in the air, an impossibility" (Maitland 1936, p. 19).

It is in the nature of equity to have the final word. Equity has long come at the end of practical reasoning in cases and controversies, pronouncing its judgments after law has finished deliberating and has rendered its final judgment. For centuries, the job of the Chancellor was to protect the crown's conscience by rendering justice when law was not up to the job. But law always had the first opportunity to set the matter right. To this day, equity will not act if legal remedies are adequate to make a wronged claimant whole (Kronman 1978, pp. 355–58; Rendleman 1981, p. 346). Though the inadequacy requirement has come under scholarly criticism in recent decades, courts continue to affirm and obey it (Bray 2015, pp. 1007–8, 1026–28).

That equity comes after law is what makes equity both useful and dangerous. Equity's final word is the source of the dilemma that equity is required to complete the law, but equitable discretion threatens to nullify law's reasoned judgments (Rundell 1958, pp. 75–76; MacLeod 2018b, pp. 124–25). The nature of law is to be universal. The language of law must speak in general terms. And "human foresight" cannot anticipate every single case in which a legal rule might have application (Home 2014, p. 25). These limitations prevent

law from accounting justly for every single instance in which the purposes of a law may be at stake.

As Story explained in his landmark *Commentaries* on equity, "It is impossible, that any code, however minute and particular, should embrace, or provide for the infinite variety of human affairs, or should furnish rules applicable to all of them". Consequently, "cases must occur, to which the antecedent rules cannot be applied without injustice, or to which they cannot be applied at all". The words of law being necessarily general and universal, they must embrace all cases, but there will arise some cases that "could not have been intentionally embraced", consistent with natural reason (Story 1836, pp. 8–9).

Aristotle taught that the nature of equity is to correct the law when it is deficient as a result of its generality. It is impossible to lay down a general, universal law to resolve all questions as the legislator would have intended, consistent with the purpose of the law, so in many cases, a decree is needed (Aristotle 1973, p. 119). But having corrected the deficiency resulting from law's generality, equity threatens to weaken the rule of law. The problem, Aristotle explained, is that equity corrects the law according to the requirements of natural justice, not legal justice (Aristotle 1973, p. 118).

As the jurists have long recognized, the power to render a decree is the power to undo the law. If a court of equity possessed the unbounded power of "superseding the law" and rendering judgment according to the dictates of the Chancellor's conscience, then "it would be the most gigantic in its sway, and the most formidable instrument of arbitrary power, that could well be devised. It would literally place the whole rights and property of the community under the arbitrary will of the judge" (Story 1836, p. 16). Equity would then "rise above all law" and become "a most arbitrary legislator" (Blackstone 2016, p. 284). It was this dangerous tendency of equity that motivated John Selden's famous, sardonic remark that equity "is a roguish thing" measured by "the Chancellor's foot" (Rundell 1958, pp. 71–72).

Even equity's most ardent cheerleaders, such as Henry Home Lord Kames, have recognized the danger here. To do the job well, a judge sitting in equity must have discretion to decree equitably. But our judges are not angels; "men are liable to prejudice and error, and for that reason cannot safely be trusted with unlimited powers". A decree that is not governed by general rules "will often be arbitrary, and substantially unjust" (Home 2014, p. 27).

In Anglo-American law, jurists solved the problem by constraining equity within clearly defined jurisdictional bounds and by bounding judicial discretion within the common-law jurist's favorite source of power and limitation: precedent. Henry Smith has neatly summarized the development this way:

> Equity courts would not change the law, but they could prevent people from enforcing legal judgments that were inequitable. So, for example, if someone accepted payment on a debt and promised not to sue, but the debtor did not secure the formality of a cancellation of the debt, the equity court would enjoin an action at law or a judgment on the debt as against conscience. Just this kind of intervention eventually led to a showdown between the law courts and the Chancery--pitting Edward Coke against Thomas Edgerton (Lord Ellesmere) and Francis Bacon--in the early seventeenth century. While equity ultimately prevailed, the equity courts developed doctrines of self-restraint to prevent further backlash. (Smith 2021, p. 1060)

The result, as one influential scholar of equity described it, was an equitable jurisdiction that was subsidiary and obedient to law, recognizing the common law as binding upon it (Rundell 1958, p. 82).

## 3. Law Is the Practical Reasoning We Carry Out Together

Equity's potential to undo law matters because law is the practical reasoning that we carry out together. Law supplies the vernacular of our civic discourse, the structure of our civic institutions and the conclusive reasons each of us has to act justly toward each other

person. Law is thus a "paradigm of public reason and choice" (Finnis 2011a, p. 233). Equity therefore has strong reasons to prescind where its intervention would abrogate or injure the law.

While different theories view law from different perspectives, the point of view that is internal to the law-abiding person's own deliberations and actions yields the insight that law is, before all else, a special kind of practical reasoning (Finnis 2011b, pp. 23–45; 2011c, pp. 3–19; Hart 2012, pp. 82–91). Legal reasoning is a participation in practical reasoning generally, though many of its reasons are artificial and technical (Finnis 2011a, pp. 18–40, 212–30). As Joseph Raz has observed, "all practical conflicts conform to one logical pattern: conflicts of reasons are resolved by the relative weight or strength of the conflicting reasons which determines which of them overrides the other" (Raz 1999, p. 35). Legal reasons supply particularly conclusive, peremptory reasons for action which override and exclude from deliberation various basic, first-order reasons that one might otherwise have to act or refrain from acting in a particular way (Raz 1999, p. 44). For example, at any moment, a person may have various reasons to honor a promise and reasons not to honor the same promise. But if the promise is binding, then the promise itself provides a conclusive reason to act, notwithstanding all the other reasons for and against performance. When equity steps in, it must account for the conclusive reason that law supplies.

To be sure, legal reasons differ in important respects from other conclusive or exclusionary reasons because law has essential characteristics that differ from the characteristics of other, normative orders (Finnis 2011a, pp. 219–20; Raz 1999, pp. 149–76). But legal reasons function within practical reasoning the same way that other second-order reasons do: as sound reasons for action that conclude practical deliberation about what to do (Finnis 2011b, pp. 28–32, 91, 103–6). Law resolves practical questions when the other reasons for or against a particular action conflict. Whether it rests in a community's agreement, stipulation, assent or other means of authoritative settlement, a legal reason—a right, duty, remedy, wrong, etc.—is a strong, second-order reason that we will act for *this* reason and not *some other* reasons that might otherwise have motivated our actions in this or that case, or (more usually) in general.

Not all legal reasons are equally strong, conclusive or exclusionary of other reasons. But we can look at central instances in which we most clearly perceive the function of legal reasons to resolve practical questions and then map weaker legal reasons relative to those (Finnis 2011c, pp. 9–11; MacLeod 2015, pp. 173–215; 2018a, pp. 295–307). In answer to the practical questions, *should I take this thing that does not belong to me? Should I flout the rules of my professional association?* And *should I falsely suggest that my competitor cheated her clients?* The law answers conclusively: *No*. To the practical questions, *should I perform this promise that I made despite a better opportunity elsewhere? Should I pay my dues?* And *should I ask permission before using her things?* The law answers conclusively: Yes. Whatever otherwise-sound reasons we might have to answer those questions differently in different circumstances, law supplies conclusive reasons to act consistently with our duties toward others and thus to stay within a community's settled practical reasons.

This practical orientation is true of all law. But it is easier to see of some laws than of others. The community's practical reason within the practice of criminal law is usually overshadowed by the coercive powers of the institutions of criminal enforcement. People pay more attention to the police officer than to the speed limit. When a criminal jury pronounces its verdict, not everyone perceives that twelve of their fellow citizens have weighed the evidence, assessed the credibility of the witnesses, listened to the judge's instructions of the laws that apply to all citizens alike and reasoned together about the question whether the accused committed an act of public wrongdoing as defined by the laws. Many people only see the handcuffs and the prison bars.

The ability of law to facilitate practical reasoning in a community is no easier to perceive in constitutional law. Constitutions are not simple things. Constitutional law is complicated by its ambition to empower public officials to alter and settle the legal rights and duties of persons and citizens while also constraining those same officials within legal

boundaries. And most constitutions leave most practical questions unanswered. Our ideas about public and constitutional law are largely determined by our prior ideas about "what forms of official conduct ought to be controlled by law" (Endicott 2003, p. 86), and different political communities may answer that question differently.

To see clearly how law enables us to reason together, it is therefore fruitful to focus first on private law rather than to begin with public law. Private law has indispensable instrumental value. Private rights and institutions enable us to coordinate so that we can survive, create, learn, nurture our children and form societies and associations for the sake of religious exercise, friendship, professional achievements and many other inherently worthwhile goals. Private law is the law that a community makes to coordinate our actions within that community for a common good. It is good to have a sound account of private law before one attempts to reason equitably so that one does not, in the exercise of equitable discretion, mistakenly undo the fruitful practical reason that people have already performed.

It is also best to begin our inquiry with private laws that work well. Unfortunately, law schools train lawyers to attend to those cases in which legal reasoning has *failed*, where the practical reasoning of one or both parties broke down and where coercion is necessary to provide a remedy or sanction for the failure. But coercion and sanction are auxiliary reasons to obey the law; they are not legal reasons themselves (Raz 1999, pp. 161–62).

In practice, many lawyers mostly deal with cases in which coercion is unnecessary because private law has succeeded in facilitating two or more people in reasoning together and coordinating their actions. And most of the time, lawyers are completely unnecessary. Most private law governs our daily deliberations and actions without the need for coercion and without the intrusion of complex constitutional institutions. Each day, millions of people walk past others' property without injuring it, perform promises on which others have relied, adhere to bylaws and organization policies, repair defective premises at the request of their tenants, pay dividends to their shareholders and lawful wages to their employees and perform countless other lawful acts in obedience to the conclusive reasons that private law places within their deliberations.

The rights and institutions of property, contracts, torts, trusts, commercial transactions, admiralty and maritime, and other parts of private law are artificial in the sense that they are artifacts of human deliberation and action. But they are not arbitrary. In its central cases, private law is not a product of power relations and discursive regimes, as many scholars now mistakenly assert. The exploitation of one person by another is precisely an instance of *defective* private law. The core meaning of private law refers to the effective, central cases in which law informs and directs the deliberations of two or more persons to coordinate their actions by particular acts or omissions so that one or more of them can achieve some rationally desirable end.

We need private law in our associations and institutions because reasoning about what is good and right to do is often difficult. Sound practical reasoning is difficult enough when one person reasons alone about his own individual goals. Most of the options that lie before us in any given moment are incommensurable, so we must choose one course of action amongst reasonable alternatives (Finnis 2011c, pp. 92–95, 114–18; Raz 1986, pp. 321–67). To choose one goal amongst various alternatives and to plan effective means to that end that are not likely to be too costly or burdensome is an exercise that requires knowledge, wisdom and prudence. In addition, to be a just person is also to ensure that both the means and the end are right, or at least not inherently wrong—consistent with one's other commitments and one's whole plan of life, not unjust toward oneself and one's family and friends, not arbitrarily partial to one's own interests, not unlikely to succeed and not untruthful about the facts of the world and the order of human conventions and society (Finnis 2011c, pp. 100–33).

One feels the burden of practical reasoning acutely when making major decisions that alter large portions of one's life. To choose a career or to decide whether to accept a job offer is to be aware that practical reasoning is not easy and that there are many ways to go

wrong. But the same is true on a smaller scale of the practical reasoning we do every day. To work late or instead attend a child's sporting event is a choice that may have lasting effects either way. We may not be able to assess those effects in advance. Indeed, we may not even be aware of what the effects are likely to be. Even when we are aware of what is at stake, it can be difficult to know what is right to do.

Conclusive second-order reasons, such as those supplied by law, enable us to overcome many of the obstacles that lie between a deliberating person and a reasonable course of action. Making decisions and plans for oneself requires accounting for all of the various goods that are wrapped up in one's own life. It requires forming and executing a plan to bring them into some semblance of order. It is the function of law to order human actions by ruling some out of consideration and presenting other actions as most consistent with a good life lived as a whole.

Consider a promise. To keep it simple, consider a promise that one makes to oneself, rather than to another person. A promise to oneself is a law that one promulgates for oneself. Once made, the promise orders one's choices and actions and makes it unnecessary to deliberate from scratch every time one faces a potential conflict between valuable opportunities. This feature of promising enables us to build lives of flourishing and personal autonomy, in which we impose obligations on ourselves by our own free choices for our own purposes (Fried 2015, pp. 1, 57).

Suppose one has promised oneself to finish a degree program at a university. That promise, freely chosen at one moment, acts as a binding obligation on oneself at subsequent moments and thus shapes one's life, ordering one's later deliberations, choices and actions (Raz 1986, pp. 385–90). Whatever reasons or other motivations one may later have to abandon one's studies, the promise overrides those motivations (absent a new, especially weighty reason, such as dire necessity) and excludes them from one's reasonable deliberations (Raz 1986, p. 388). One may be tempted to slacken one's efforts. Or one may later discover other valuable goals to pursue. One may encounter strong reasons to spend time on other pursuits, such as invitations to join friends on a long trip or an attractive job offer. But to allow those reasons to deter one from the goal of finishing the degree program would be to let oneself down, to act unjustly toward oneself. Once made, the promise excludes from reasonable consideration all of the reasons and other motivations that might disorder and impede one's pursuit of the degree.

To reason practically in community with other persons—other choosing, moral agents—is far more complicated than to reason alone. To make decisions and plans for two or more persons is to multiply the goods at stake and the potential for conflicts between them. Even two people who want to cooperate with each other to achieve a common end and reason together in good faith may find cooperation difficult. Each of them comes to the common enterprise with a complex variety of prior commitments and priorities. The goods wrapped up in each person's life plan must, to a greater or lesser extent, be re-ordered to make room for the goods at stake in the common enterprise. How to reorder them is often a difficult question. There is seldom a uniquely correct answer. The incommensurability of human goods and plans does not dissolve when one multiplies the goods and plans at stake (Finnis 2011a, pp. 234–43). To the contrary, the problems of commensuration multiply (Raz 1986, pp. 340–53).

Even if we can somehow manage to identify and reconcile the goods at stake, practical reasoning with others runs into additional obstacles. First, one must attend to one's own ignorance of the plans of the other person. And one must take one's ignorance of those plans into account. This attentiveness to the plans of others is a requirement of justice. We can never be justified in factoring other people into our own plans as mere means rather than ends in themselves. Each human being is a being with intrinsic worth, an independent agent of practical reason. So we must ask questions, listen carefully and try to understand the reasoning of other people with whom it is sometimes our joy, and sometimes merely our obligation, to coordinate our plans and actions.

Second, we must attend to our own ignorance of the world around us. Even if we are fortunate enough to live within functioning market economies whose prices signal to us effectively the instrumental value of goods and services, even if we are well educated in the basic facts and methods of inquiry concerning nature and the artifacts and conventions within our society, and even if we understand the language and other signifiers of meaning that our interlocutors use to communicate—those are all contingent "if"s—we still often know far less than we think we know.

Third, we all go wrong in our practical reasoning from time to time. Different plans are often incommensurable, but some plans are simply wrong, even unjust; the reasonable is commensurable with and superior to the unreasonable (Finnis 2011a, pp. 235, 243–55). We must account for the possibility that our own understanding is clouded and our own reasoning is corrupted by motivations that are less than pure. And, as difficult as it is to understand one's own sub-rational motivations, it is often nearly impossible to understand what motivates another person to act selfishly or unjustly and to anticipate when they may do so again.

Even if each of us had perfect knowledge and pure motivations, each person still confronts the problem of deciding with another person what to do and how best to do it. Just as any individual's decision must necessarily involve choosing one particular good and rejecting other, alternative goods that might have been pursued, a decision by a group of people to pursue some desirable end together just is the rejection of many alternatives. Some or all of those alternatives may have been as worthwhile and fully reasonable as the one chosen. And each member of the group may have a different and equally reasonable proposal than all the others. Our own preferences and biases may prevent us from seeing the true worth of the various options, just as the biases and preferences of those with whom we deliberate may shield from their view the value of the option that we prefer.

Practical reasoning becomes more complicated as we involve more people in our cooperative plans. The difficulties multiply as the enterprise is scaled up. We cannot achieve anything at all unless we have some way to settle upon some goal and plan of action to the exclusion of other possible goals and plans. We must stick with our goal and plan long enough for the plan to succeed. And we must account for the failures and frustrations that we will experience along the way.

Law responds to those complications. When equity steps in to provide a remedy for some break-down in practical reason, it should do so primarily by reference to how law works when it works well. Otherwise, equity might destroy the best tool we have to perform practical reasoning together. To reason with others is difficult. A person reasoning equitably should bear in mind that, when sound and just, laws solve the problems of practical reasoning in community.

## 4. Law Gives Us Particular Reasons to Act for the Common Good

Law is not a perfect solution to all of those problems. But it is the best solution that humans have come up with. The norms and institutions of our fundamental law enable us to reason together and to overcome the many obstacles to cooperative action that we regularly encounter as we pursue our common good. The law performs this indispensable function by identifying (and, when necessary, enforcing) our "publicly adopted reasons for adopting or rejecting proposals for action, public or private" (Finnis 2011a, p. 233).

We all have reasons to act for the common good. The law gives those reasons specific form and content to make them conclusive and binding. It specifies that some good actions are legally right and marks off some actions as legally wrong. And it supplies institutions which stand ready to provide sanction in response to legal wrongs that affect the political community as a whole and remedies in response to wrongs that infringe a particular private right.

Consider just the most basic private rights, duties and institutions. Property rights and institutions coordinate our impersonal interactions (Merrill and Smith 2007). They enable us to act well and justly toward persons we do not know. A person who refrains

from injuring other people's resources and from using and possessing those resources without permission enables worthwhile use of things insofar as he avoids interfering with the reasonable plans of action of other persons and groups (MacLeod 2015, pp. 91–121). Every time each of us walks past a family business without stealing or damaging its equipment and products, or pays royalties to an artist to obtain or use a copy of her song or painting, or refrains from barging into another family's home during meal time, we are negatively coordinating with human beings whom we may never meet to help facilitate their productive use of some resource (Claeys 2018). Each of those persons and groups is entitled to our respect as equal, rational agents of choice and action (MacLeod 2015, pp. 91–121; Dagan 2021, pp. 41–78). Property rights make this coordination possible by providing what practical reason requires. Property gives specific, context-independent content to the general obligation of abstention that we all owe to others not to interfere in their productive uses (Penner 1997, pp. 128–52; Merrill and Smith 2007, pp. 1870–90; Claeys 2017).

Whether a contract is understood as an act of promise or one of consent, contract law enables us to form and structure our personal, exchange and commercial interactions with other persons in the ways we think are most likely to help us achieve our common good ends (Raz 1986, pp. 82–84, 173–76; Barnett 1986, pp. 294–300; Barnett 2010, pp. xii–xxi; Finnis 2011c, pp. 139–40, 232; Fried 2015). By promising according to the rules of contract law, one imposes duties on oneself and confers correlative rights on others, creating new normative bonds between oneself and others (Raz 1986, pp. 83, 173–76; Hart 2012, p. 43). By consenting or assenting to another's promised offer, one alters one's normative status in a similar (though not identical) way, assuming and imposing certain duties and conferring and accepting correlative rights (Raz 1986, pp. 82–84).

Bilateral contracts and the formal requirements of contract law that signal how to make such contracts enforceable empower two people to deliberate about and make binding promises to each other. That power in turn enables each contracting party to change his legal relations with respect to each other and to act in reliance upon the necessary, coordinated actions of the other (Fuller 1941). Contracts also enable us to sequence the performances of our promises, using terms and conditions expressed in a shared technical language (Blum and Bushaw 2022, pp. 697–729). Well-crafted contracts, written to comply with sound legal doctrines, help us to achieve common ends while also satisfying the requirements of justice that we not defraud or exploit each other (Barnett 1986, pp. 299–300). The boundaries that the law places around enforceable contracts, such as doctrines of fraud, duress and illegality, ensure, to the extent possible, that contracts are not used as instruments of private wrongdoing (Barnett 2010, pp. 209–40).

The doctrines of remedies and sanctions—torts, criminal law and equitable remedies—account for the ways in which our practical reasoning most often fails. When a person intentionally and culpably commits a wrong against the public, the act of injustice cannot be undone, but law can restore some measure of justice (Finnis 2011c, pp. 263–64). A criminal sanction can provide some measure of retributive justice, which is usually the closest thing we can get to justice against a person who has placed himself above the law (Bradley 2003). Similarly, damages and other remedies for private wrongs cannot always put the injured party in exactly the condition that she should have occupied absent the trespass, negligence, or breach. But courts can often administer remedies that fit the injury more or less (Klimchuk 2001; Goldberg and Zipursky 2010), and a remedy that fits more or less is better than no remedy at all (Ripstein 2015).

Private law is pluralist, like the practical reasoning that it facilitates. We all do and make private law together with others. But we do not do it all together. Private law is the practical reasoning that we do in those various, overlapping groups and associations that comprise any healthy, functioning society. A family and a family business; a church, a synagogue, and a mosque; a school, college and university; a fraternal society, a defense auxiliary and a volunteer fire brigade; a neighborhood association; two businesses contracting for the manufacture and sale of goods; two people contracting for the sale of an antique

chair—all of these little political communities make and obey their own laws. Without forcing their laws on anyone else, they flourish precisely to the extent that they reason and act together according to their laws.

To read the law is to see that little of the law which governs our daily lives is made and enforced by governments. Most of it we make ourselves, acting alone, in contracting pairs, in meetings of boards, faculties and officers and in countless other acts of collaborative, practical reasoning. Even the arms-length contract only incidentally implicates sovereign power. Courts may have the power to give a remedy in the event of a breach, but only if the aggrieved party can make particular proofs in a court of law or equity. What we call "contract law" is nothing like the sum of the law that governs contracts. It is instead a gatekeeper to keep most promises *out* of court; it is the law that determines which promises may be enforced in courts of law and equity and which may only be disposed of in foro conscientae. Most promises are left to the latter (Story 1836, pp. 2–3). That is not the only Latin term for the government's abstention from unsettling legal acts. At common law and in equity, many acts of wrongdoing are damnum absque injuria, injuries without any damages or remedy (Story 1836, p. 12).

Because private law serves so many indispensable functions in our cooperative reasoning, equity does well by leaving the rights, duties and remedies of private law in place. Political regimes that empower judges and other powers to act on behalf of equity in disregard of the law exhibit all of the flaws of sovereign command regimes but none of the benefits. When those entrusted with the power to administer equity supersede the settled, practical reasons of the law, equity's judgments do violence to the common good that our plural communities set out to achieve. Equitable judgments that supersede law are just as arbitrary as assemblages of signs that indicate a sovereign will. And unlike the public declaration of a stable, sovereign volition, equitable pronouncements aiming at equal results, in particular judgments rendered case by case, lack even the virtue of stability. A community cannot plan around a future assessment of likely consequences. This is especially true when the assessment is to be made by some person who is not personally invested in the goods and plans of the groups and associations whose rights and institutions are at stake.

Equity rejects as foolish the aspiration to replace the law. To monopolize control over the law would be to destroy communal practical reasoning. As long as human beings are reasoning animals, reasons for action will always be in high demand. Monopolies always create shortages of supply.

## 5. True Equity Models Civic Discourse

In its most general and widely accepted form, equity is the charitable reading of law which completes the law and maintains the integrity of law toward its own purposes. For this reason, "Chancery is ordained to supply the law and not to subvert the law" (Story 1836, p. 13). Equity must be exercised "not according to men's wills and private affections" but according to the rules of "law and equity", which do not contradict each other but complement each other (Story 1836, pp. 15–16).

As Aristotle taught, because the law must speak generally and universally, equity must sometimes attribute to the legislator what the legislator would have said about a particular case and read the law as containing that ruling (Aristotle 1973, pp. 118–19). English and American jurists referred to equity's charitable interpretation and application of law as equity's "true and genuine meaning (Blackstone 2016, p. 282; Story 1836, p. 7)". As Story described it, equity in its most distinct form, "as contradistingished from mere law, or *strictum jus*", is the interpretation of the written words of law "by construing them, not according to the letter, but to the reason and spirit of them" (Story 1836, p. 7). This sense of law places equity not in tension with law but rather in cooperation with its most fundamental reasons.

At its core equity is, as Blackstone called it, "the soul and spirit of all law", which construes positive law according to its basic reasons and thereby produces "*rational* law"

(Blackstone 2016, p. 282). Equity constrains courts to assume the best of legislatures and other law makers rather than the worst. It operates within a strong presumption that, had the lawmaker been confronted with a potentially irrational or unjust application of the law, then the lawmaker would have said, *No. That is not what I intended*.

### 6. Conclusions: Equity and Civic Discourse

The functions of equity in adjudication point toward a model of equity in practical reasoning and civic discourse more broadly. Equity teaches us how to assume the best of each other rather than the worst. An equitable interpretation of the law teaches us how to understand each other charitably, starting with a presumption of reasonableness rather than malice or wrongful discrimination. Equitable administration of justice points us to the first principles of natural justice, without attention to which practical reasoning often runs contrary to the common good. And just as equity defers to law in adjudication, an equitable mode of civil discourse would defer to those settled customs and conventions that reflect our communal deliberations and judgments.

We do especially well to bear this last feature of equity in mind today. Above all, true equity causes no harm to law. Law is our shared practical reasoning, the reasoning we carry out together to solve our practical problems. Equity's respect for law is a model for equity's more general aspiration to complete and perfect public discourse about important civic questions, the first imperative of productive civic engagement. Educators in law and related disciplines would do well to recover the tradition of teaching law and equity together.

At a minimum, law schools would do well to restore jurisprudence to its privileged place in the curriculum. Students should be enabled to see that some practical questions are universal and that law and equity are different tools to answer practical questions. This will help students perceive what law and equity have in common and why both are important. Beyond that, equitable interpretation of the law might be woven into the study and interpretation of laws, as Aristotle contemplated. To connect equity to the study of law is to perceive the common object that animates both law and equity, namely that we achieve justice in our communities as we reason together about what is to be done.

**Funding:** This research received no external funding.

**Institutional Review Board Statement:** Not applicable.

**Informed Consent Statement:** Not applicable.

**Data Availability Statement:** No new data were created or analyzed in this study. Data sharing is not applicable to this article.

**Conflicts of Interest:** The author declares no conflict of interest.

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
