# Peer review of "Why Equity Follows the Law"

_laws, 2023_

Round 1

Reviewer 1 Report

Comments and Suggestions for Authors

Excellent piece of work. All the arguments are well justified, structured and presented in a very convincing and logical way. The language and style are clear, complicated enough for this type of scholarly work but not way too complex. The conclusion is very logical; differences between the values of equity and law are well understood by the author while the interaction between these two key concepts (or how it should go) is professionally described / laid out. The research for the paper is done at a a very high level; all the notable and/or prominent scholarly authorities on legal theory and philosophy of law are properly ananysed and subsequently cited. All in all, a great and exemplary piece of academic work which I recommend for publishing without any reservations. Kudos to the author!

Author Response

Thank you for reading it and for your charitable comments.

Reviewer 2 Report

Comments and Suggestions for Authors

Excellent piece! I feel that I have been schooled--in the sense that so many pieces of a puzzle that I know have been masterfully assembled before me and applied to a current problem in record word count with insanely thorough citations to varied, canonical, and rock solid sources. The work itself is a testament to the power of practical reason: it beautifully illustrates how genuine understanding of our legal tradition guides and trains one to apply that tradition--in highly practical and reasonable ways--to issues as they emerge (or reformulate) over time. Lovely and needed application of old insights to new problems.

A very minor suggestion: the return to the subject of equity near the bottom of page 9 was a bit jarring. Would it be possible in pages 3-9 to either forecast or remind the reader of the return to equity that is coming? The argument that leads up to this point is beautifully crafted, but reminding the reader that it will lead back to explaining the proper role of equity in civil discourse in some way might help the whole tie together more smoothly--particularly for any reader's less well versed in the sources being drawn upon. 

Author Response

Thank you for the charitable reading of, and comments on, my article. Your suggestion is excellent and well received. I will make that change.

Reviewer 3 Report

Comments and Suggestions for Authors

This is an interesting, timely, and learned piece. It draws from both recent and classic scholarship to make its points. And it is written in a clear, easy-to-follow way. I have no strong objections to the article. I do have two suggestions for improvement.

The first suggestion is to offer a definition of "equity" early in the paper. The essay begins with a section called "essential characteristics of equity." And scattered throughout are descriptions or appraisals of equity, such as Blackstone's ("the soul and spirit of all law"). But I would like to see the author define this term in her or his own words. 

The second suggestion is to flesh out a bit more fully what a curriculum that takes equity seriously would look like. The introduction to the essay argues that "law schools have a particularly important role to play in teaching equity and law together." And the concluding words are that "educators in law and related disciplines would do well to recover the tradition of teaching equity and law together." Even a few sentences to lay out how this might be done would be welcome. (Should this be the focus solely of jurisprudence/philosophy of law classes? Or should considerations of equity be part of the entire legal curriculum?) 

Author Response

Thank you. Both good suggestions that I will take on board. I am loathe to define equity definitively. I think a focal meaning is better suited to the concept, which lacks precision. But I will provide a working definition by reference to law.